# Identifying Novel Osteoarthritis-Associated Genes in Human Cartilage Using a Systematic Meta-Analysis and a Multi-Source Integrated Network

**DOI:** 10.3390/ijms23084395

**Published:** 2022-04-15

**Authors:** Emily Shorter, Roberto Avelar, Margarita Zachariou, George M. Spyrou, Priyanka Raina, Aibek Smagul, Yalda Ashraf Kharaz, Mandy Peffers, Kasia Goljanek-Whysall, João Pedro de Magalhães, Blandine Poulet

**Affiliations:** 1Department of Musculoskeletal and Ageing Science, Institute of Life Course and Medical Sciences, University of Liverpool, Liverpool L7 8TX, UK; r.avelar@liverpool.ac.uk (R.A.); priyankaraina10@gmail.com (P.R.); aibek@liverpool.ac.uk (A.S.); yalda@liverpool.ac.uk (Y.A.K.); peffs@liverpool.ac.uk (M.P.); kwhysall@liverpool.ac.uk (K.G.-W.); aging@liverpool.ac.uk (J.P.d.M.); bpoulet@liverpool.ac.uk (B.P.); 2Bioinformatics Department, The Cyprus Institute of Neurology & Genetics, Nicosia 23462, Cyprus; margaritaz@cing.ac.cy (M.Z.); georges@cing.ac.cy (G.M.S.); 3Department of Physiology, School of Medicine, The Regenerative Medicine Institute (REMEDI), NUI Galway, H91 TK33 Galway, Ireland

**Keywords:** microRNA, osteoarthritis, cartilage, meta-analysis, mRNA, novel, joint, dysregulation, proteomics

## Abstract

Osteoarthritis, the most common joint disorder, is characterised by deterioration of the articular cartilage. Many studies have identified potential therapeutic targets, yet no effective treatment has been determined. The aim of this study was to identify and rank osteoarthritis-associated genes and micro-RNAs to prioritise those most integral to the disease. A systematic meta-analysis of differentially expressed mRNA and micro-RNAs in human osteoarthritic cartilage was conducted. Ingenuity pathway analysis identified cellular senescence as an enriched pathway, confirmed by a significant overlap (*p* < 0.01) with cellular senescence drivers (CellAge Database). A co-expression network was built using genes from the meta-analysis as seed nodes and combined with micro-RNA targets and SNP datasets to construct a multi-source information network. This accumulated and connected 1689 genes which were ranked based on node and edge aggregated scores. These bioinformatic analyses were confirmed at the protein level by mass spectrometry of the different zones of human osteoarthritic cartilage (superficial, middle, and deep) compared to normal controls. This analysis, and subsequent experimental confirmation, revealed five novel osteoarthritis-associated proteins (PPIB, ASS1, LHDB, TPI1, and ARPC4-TTLL3). Focusing future studies on these novel targets may lead to new therapies for osteoarthritis.

## 1. Introduction

Osteoarthritis (OA) is the most common musculoskeletal disorder and cause of chronic disability in adults [1]. The main characteristic of OA is the deterioration of the articular cartilage, of which chondrocytes are the only cell type. The primary function of chondrocytes is to maintain homeostasis of the extracellular matrix (ECM). During OA, chondrocytes show aberrant phenotypes and actively produce cartilage-degrading enzymes, such as matrix metalloproteinases (MMPs) and aggrecanases, which result in the destruction of the ECM [2]. This change in phenotype is reflected at the mRNA level with various studies having identified a number of differentially expressed genes, some of which can also be seen at the protein level [3,4,5]. Despite research having identified a number of key genes and cellular pathways associated with the pathogenesis of OA, their potential as therapeutic targets remains largely undetermined.

Micro-RNAs (miRNAs) are a class of small non-coding RNA molecules, approximately 22 nucleotides long, which bind to messenger RNAs (mRNA) and induce their degradation or inhibit protein translation. They play a major role in regulating post-transcriptional gene expression and therefore protein levels. miRNA dysregulation has been implicated in OA development [6] and they are emerging as powerful regulatory molecules and novel therapeutic agents. miRNAs often have hundreds of experimentally verified and/or predicted target genes, which can be accessed via public databases such as miRTarBase [7] and TargetScan [8] and can therefore target multiple genes involved in a specific disease process. It is thought that restoring physiological levels of miRNAs, via mimics or inhibitors, will allow for restoration of joint homeostasis and function. Although miRNA and gene expression data are incredibly useful for understanding diseases, they are primarily researched independently. Single nucleotide polymorphisms (SNPs)—a common genetic variation where single nucleotides are substituted at a specific position in the genome—are another important tool for identifying disease-causing genes [9]. The benefit of integrated biological networks is that they combine these data and allow us to gain new insights into the molecular interactions underlying diseases [10]. These networks can be used to identify potential new genes that contribute towards a biological process or disease phenotype, and aid in target prioritisation via a guilt-by-association approach [11].

The current study utilises a *p*-value-based meta-analysis of the literature to identify miRNAs and mRNAs that are significantly dysregulated in OA cartilage. Subsequent pathway enrichment, overlap, chondrocyte co-expression, and integrated network analysis using a variety of input data is then used to identify novel OA-associated hub genes, which are ranked and prioritised. It is anticipated that this research will highlight potential important therapeutic targets and pathways that future research should focus on.

## 2. Results

### 2.1. p-Value-Based Meta-Analyses Identify 6 miRNAs and 207 mRNAs Differentially Expressed in OA Cartilage

The PubMed search for studies on mRNA and miRNA expression in OA yielded 936 and 622 papers, respectively. Of these initial papers, 86 on miRNA expression and 30 on mRNA expression met our eligibility criteria (see Literature Search and Eligibility Criteria in Methods). Studies on miRNA expression were subject to quality control based on the inclusion of the miRNA in miRbase (v22; http://www.mirbase.org, accessed on 1 May 2020), resulting in 77 papers that were suitable for the meta-analysis. From these papers, 411 miRNAs and 5166 mRNAs were extracted. The *p*-value-based meta-analysis identified 6 miRNAs and 207 mRNAs as being significantly dysregulated in OA cartilage compared to healthy tissue in three or more independent studies. The 20 top mRNAs are shown below (Table 1) and the full list can be found in Appendix A.

A further 27 miRNAs were found to be differentially expressed in two or more independent studies (Appendix A).

### 2.2. Ingenuity Pathway Analysis (IPA) of Significant mRNAs and miRNA Target Genes Reveals 12 Shared Chondrocyte Pathways Linked to OA, of Which Senescence Is the Most Significant

To determine the most significant cellular pathways linked to these dysregulated mRNA and miRs, we performed pathway analysis (IPA) on the 207 dysregulated mRNAs and on the target genes of the miRNAs identified from the meta-analysis. We identified four common pathways between the dysregulated mRNA genes and miRNA target genes, including senescence, p53 signalling, BEX2 signalling, and unfolded protein response (Figure 1a). We next investigated further the most significant pathway of senescence and showed that our lists of miRNA target genes, mRNAs, and their upstream regulators overlapped with genes shown to induce or inhibit senescence in vitro (CellAge genes) [12]. The most significant overlap was between predicted upstream regulators of the differentially expressed mRNAs identified in the meta-analysis and inducers of senescence (43% overlap). Moreover, there was a 33.3% overlap between inhibitors of senescence and predicted upstream regulators of miRNA target genes (Figure 1b). All significantly enriched canonical pathways identified by IPA in the list of significant miRNA target genes and mRNAs can be found in Appendix A, respectively.

### 2.3. Analysis of the Multi-Source Information Network Identifies and Ranks OA-Associated Hub Genes, 32 of Which Are Confirmed in Mass-Spectrometry Data of OA Articular Cartilage vs. Healthy Controls

An undirected and weighted chondrocyte co-expression network was built using 396 human chondrocyte samples (see ‘Chondrocyte expression matrix construction’ in methods). This analysis allows the identification of novel genes that may not have been associated with OA previously, but that are commonly co-expressed with OA-associated genes identified from the meta-analysis. The meta-analysis genes were used as seed nodes. First-order interactors were extracted alongside the OA genes and disconnected components comprising less than 10 nodes were filtered out. The resulting network contained 1463 unique nodes and 4972 edges shared across seven disconnected components (including 142 OA seed nodes).

An integrated multi-source information (MI) network was then created by combining the co-expression network with other data sources, such as miRNA target genes and SNPs. In order to condense the topological information of the network to a node (gene)-specific summary, the weighted degree was calculated, also known as the strength of each node, which was taken as the MIGe for the Multi-source Information Gain (MIG) score and was combined with the normalized integrated gene-specific information MIGn (see methods for more details). This allowed us to make a total gene score based on both node and edge aggregated score. The top 20 genes ranked by this score are listed in Table 2 and a list of all 1685 genes identified from the MI network can be found in Appendix A. Gene scores of CellAge genes were also significantly higher when compared to non-CellAge gene scores in the MI network (*p* = 0.0015).

These 1685 genes from the MI network were filtered for their inclusion in mass spectrometry (MS) data of dysregulated proteins in OA cartilage vs. healthy controls. Data are available via ProteomeXchange with identifier PXD029116. This proteomic analysis found 81, 46, and 29 dysregulated proteins in the superficial, middle, and deep cartilage zones, respectively. Filtering for genes identified from the MI network revealed 32 common proteins between the MS data and the MI network (Table 3). Seven mRNAs of these thirty-two proteins were identified in the original meta-analysis, whereas the rest were found via analysis of the MI network. Finally, five of the proteins (*PPIB*, *ASS1*, *LDHB*, *TPI1*, and *ARPC4*-*TTLL3*) identified from the meta-analysis, that were experimentally confirmed via inclusion in the proteomics data, have never been studied in OA before.

## 3. Discussion

Following a systematic literature search and data extraction, this study included a *p*-value-based meta-analysis of data from all eligible miRNA and mRNA expression studies in human OA cartilage versus healthy control tissue. We identified a list of OA-associated genes and miRNAs, some of which were also confirmed to be modified at the protein level. Using this MI network approach, a ranked system was established in order to prioritise genes that future research can study for potential therapeutic targeting. In addition, this study has identified novel OA-associated genes that were not found previously.

As senescence was one of the most significantly OA-associated pathways shared between both the miRNA target genes and mRNAs, overlap analysis was performed with genes included in the CellAge database [12]. These CellAge genes were compiled by a systematic literature search of genetic manipulation studies whereby direct in vitro manipulation of the gene in question resulted in induction or inhibition of cellular senescence. Results of this analysis revealed highly significant overlaps, the most significant being with predicted upstream inhibitors of the differentially expressed mRNAs and inducers of cellular senescence. Furthermore, CellAge genes were found to score higher in the MI network compared to non-CellAge genes, further highlighting the potential significance of cellular senescence in the development or progression of OA. Senescent cells accumulate later in life and at sites of age-related pathologies, where they contribute to disease onset and progression through complex cell autonomous and non-autonomous effects [13]. Previous research has shown that senescent chondrocytes not only accumulate with age but are present at higher numbers in human OA cartilage compared with age-matched healthy controls [14]. In fact, a clinical trial investigating whether the senolytic supplement fisetinA reduces OA-associated pain and cartilage breakdown is due to begin in 2022 (NCT04770064). A key characteristic that distinguishes senescent cells from other cell types is the upregulation of a combination of factors known as the ‘senescence-associated secretory phenotype’ (SASP) [15]. The SASP contributes to fuel a state of chronic, systemic, low-grade inflammation, known as ‘inflammaging’, and compromises a subset of genes whose encoded secreted proteins include proinflammatory cytokines and chemokines, growth factors, and proteases that can digest the ECM [16]. Overall, results of the overlap analyses corroborate this research, suggesting a strong association between OA and cellular senescence.

One of the strengths of this study is that it increased the sample size by combining all eligible data into one statistical test. This is particularly important as sample sizes of individual miRNA studies are often small, especially as control healthy cartilage is notoriously difficult to obtain. A co-expression network allows identification of genes that tend to show a coordinated expression pattern across a group of samples, in this case chondrocytes. However, hub-gene identification via co-expression networks has limited power for identifying targets for follow-up studies [11]. We therefore enhanced identification of important hub genes by integrated network analyses where additional data sources were integrated to make a multi-source information (MI) network, with the aim of prioritising genes on their importance to OA [10]. Finally, we confirmed our list of OA-associated genes at the protein level using mass spectrometry data. It is known that articular cartilage can be separated into distinct zones, namely, superficial, intermediate, and deep, in which chondrocytes show distinct gene expression profiles and behaviours [17]. Thus, separating these zones for proteomics analysis ensures location-specific changes in protein levels is not under-represented compared to whole-cartilage samples.

Using this complex method, we have confirmed some well-studied OA-associated genes, such as *TIMP-4* (Tissue Inhibitor for Matrix Metalloproteinases 4), a potent Matrix Metalloproteinases (*MMP*) inhibitor known to be expressed by chondrocytes [18]. Although not as well-studied in OA as other *TIMP* family members, previous research has demonstrated an increase in *TIMP-4* in human knee synovium and expression in primary hip and knee chondrocytes [19]. In addition, other commonly associated gene expression changes were defined by our network analysis, such as aggrecan (*ACAN*), collagens 1 and 12, and lubricin (*PRG4*) [20]. As well as confirming well-studied OA genes, results of the MI network found a few genes that have, to our knowledge, never been implicated in OA previously (*PPIB*, *ASS1*, *LDHB*, *TPI1*, and *ARPC4-TTLL3*). This shows that there may be many genes integral to OA pathogenesis that have yet to be identified in the tissue, let alone studied. These novel genes warrant further investigation to understand the complex disease more fully. Interestingly, *PPIB* has been implicated in the rare skeletal disorder ‘osteogenesis imperfecta’ (OI), with a recent study reporting a rare pedigree with an autosomal recessive OI caused by two novel *PPIB* mutations [21]. Moreover, lactate dehydrogenase (*LDH*) catalyses the interconversion of pyruvate and lactate, which are critical fuel metabolites of skeletal muscle. Previous research found that *LDHB* expression is induced by exercise in human muscle and that chronic activation of *LDHB* in skeletal muscle triggers an adaptive oxidative muscle transformation, leading to increased exercise capacity in muscle-specific *LDHB* transgenic murine models [22]. This is particularly interesting as research is increasingly finding a relationship between OA and skeletal muscle wasting [23]. This warrants further investigation into *LDHB* in both OA and surrounding peri-articular muscles. *TPI1* is another novel gene identified in this study as having a critical role in skeletal muscle, where it is involved in oxidative pathways [24]. *ASS1* encodes a protein that catalyses the penultimate step of the arginine biosynthetic pathway, mutations in which cause the life-threatening condition Citrullinemia [25], but has not, to our knowledge, been implicated in diseases of the musculoskeletal system. Finally, *ARPC4*-*TTLL3* functions as the actin-binding component of the Arp2/3 complex [26]. The fact that three of the five novel genes identified from this study are implicated in skeletal muscle function corroborates the idea that health of the peri-articular skeletal muscles may play an integral role in OA pathogenesis.

Other genes identified by the network analysis have been previously linked to OA, but have not been studied for their potential as biomarkers/therapeutic targets. For example, Apolipoprotein D (*APOD*) was found to be downregulated in every zone of the OA cartilage. It was also 1 of the 207 mRNAs identified as significantly dysregulated from the meta-analysis and was the top-ranked gene of the MI network. Research has previously implicated *APOD* as being an important gene in OA pathogenesis. For example, *APOD* is strongly upregulated by retinoic acid [27], which is in turn regulated by *ALDH1A2*—an OA risk locus [28]. In vitro studies have shown *APOD* to be upregulated upon SOX9 overexpression, a master transcription factor essential for cartilage ECM formation [29]. Furthermore, a recent study into the identification of knee OA genes shared by both cartilage and synovial tissue proposed that *APOD* may manage OA through chondrogenesis in articular cartilage and immune regulation in the synovium [30]. The high ranking of *APOD* in the MI network makes it an interesting candidate for future studies into OA. In particular, research should investigate its potential as a biomarker/drug target. Fibrillin-1 (*FBN1*) was another gene that was discovered from analysis of the MI network whose encoded protein was also found to be upregulated in the middle and deep zones of OA cartilage according to the MS data. This is particularly interesting as *FBN1* is the causative gene of the inherited connective tissue disorder Marfan syndrome [31]. Moreover, it was 1 of 300 proteins identified via lectin-affinity chromatography in a previous study investigating the proteome of human OA synovial fluid [32]. The fact that this gene encodes microfibrils that play a structural role in all connective tissues, and mutations in which are known to cause a disease of the musculoskeletal system, warrants further investigation of its role in OA.

It should be noted that different methods of RNA extraction, miRNA expression measurements, and statistical methods were not considered in this analysis. Hypothetically, the impact of these variables could be investigated systematically, for example, by performing sensitivity or meta-regression analyses. However, the current number of independent studies is too small to allow for this kind of analysis. Most of the studies used in this analysis also did not report specific *p*-values in relation to the miRNA dysregulation. Rather, the values were reported as “less than” a certain significance level (typically <0.05 or <0.001). In these instances, the largest possible *p*-value was used (i.e., if it was reported at <0.05, the *p*-value used in the analysis was 0.05). This conservative method may have prevented some miRNAs that were on the verge of significance from being included in the study. Although the sample size of analysis for each miRNA was increased by the meta-analysis method used, ultimately the quality of the analysis is only as strong as the original publications. As mentioned in the methods, quality control was conducted whereby some miRNAs and publications were filtered out based on certain criteria. However, errors or limitations of analysis in the original publications may remain. Moreover, research has suggested that there are reporting biases of differential gene expression in literature, including: preferential reporting of overexpressed rather than underexpressed genes as well as genes that are popular in the biomedical literature at large [33]. As such, a critical mRNA that is investigated by only one group worldwide may not make the cut in the present analysis despite its potential importance to the disease pathogenesis. This bias is evident in the results of this study. For example, miR-140 is probably the most researched and established miRNA to date in terms of its relation to OA [34,35]. As its dysregulation has been very well classified, research will often include it as a positive control. This is reflected in the results of this meta-analysis, where miR-140, miR-140-3p, and miR-140-5p were all found to be significantly dysregulated. However, miR-140 has also been shown to attenuate OA progression via the inhibition of senescence in a recent study by [36]. This provides further support for the downregulation of the miRNA observed in this meta-analysis, as well as the association of its target genes with senescence.

A possible weakness of this study is that eligible studies often did not specify the stage of OA of the tissue donor. As many of the samples came from total joint replacements, it is assumed that a lot of the samples were from late-stage patients. Studies have shown that different stages of OA development and severity have distinct gene and miRNA expression patterns [37]. As such, the results may not adequately represent miRNA dysregulation in early-stage OA.

## 4. Materials and Methods

### 4.1. Literature Search and Eligibility Criteria

A systematic literature search for miRNA and mRNA expression studies in human OA cartilage was performed using PubMed (http://www.pubmed.gov (accessed on 1 May 2020)), applying the search terms “(microRNA OR miRNA OR miR OR micro-RNA) AND (OA OR Osteoarthritis)” for the miRNA analysis, and “(OA OR osteoarthritis) AND (mRNA OR gene) AND (expression) AND (human) AND (knee) NOT (synovium) NOT (murine) NOT (meniscus)” for the mRNA analysis. Papers were assessed for eligibility using the title, abstract, or full text, as necessary. Only articles published in peer-reviewed journals and in English were considered. Papers were not filtered for publication date and were only considered for eligibility provided they: (1) used human knee articular cartilage tissue for analysis, (2) used control cartilage from non-OA patients, and (3) provided the number of patients and significant *p*-values. A summary of eligible studies can be found in Appendix A and an overview of the study design is depicted in Figure 2.

### 4.2. Data Extraction and Quality Control

For each eligible paper, the first author’s name, year of publication, PubMed link, city/country of origin, source of specimen, number of OA and control samples, *p*-values, miRNA/mRNA names, and direction of dysregulation were extracted. For quality control, the list of the extracted miRNAs was compared to those included on miRbase (v22; http://www.mirbase.org (accessed on 1 May 2020)). Any miRNAs that were not listed on miRbase, had insufficient annotation, or corresponded with expired/non-human entries, were excluded from further analysis.

### 4.3. p-Value-Based Meta-Analyses

A *p*-value-based method was used as it enables the combination of results when effect size estimates and/or standard errors from individual studies are not freely available. Meta-analyses were performed on *p*-values and directions of effects, providing the miRNA or mRNA was identified as being significantly dysregulated in ≥3 independent studies, as previously described [38]. To do so, a customised R Studio script was used to transform *p*-values into signed z-scores using Stouffer’s method [39,40], which were then converted to positive or negative values depending on the direction of expression (R script can be found in Appendix A). Z-scores for each miRNA/mRNA were combined by calculating a weighted sum, with weights being proportional to the square root of the effective sample size of the study.

### 4.4. CellAge Overlap

CellAge is a database of genes that can drive the senescence process [41]. Build 2 of CellAge [12] was overlapped with differentially expressed genes identified from the OA meta-analysis. Significance was assessed using a two-tailed Fisher’s exact test with Benjamini–Hochberg false discovery rate (FDR) correction.

### 4.5. Chondrocyte Co-Expression Matrix and Network Construction

The undirected weighted protein-coding chondrocyte expression matrix was built using 396 chondrocyte read count data obtained from recount2 [42]. The raw expression data were normalized by quantile normalization method using Bioconductor in R Studio [43]. Protein-coding genes were obtained using Ensembl biomaRt version 101 [44]. The mutual-rank method was used to obtain the top co-expressed partners for all 15,550 protein-coding genes expressed in the chondrocyte data [45]. The mutual-rank cut-off value of 15 was used to filter the top co-expressed genes, resulting in 14,521 unique nodes and 61,222 edges. Networks were built and analysed using the R package igraph version 1.2.521 [46]. The chondrocyte co-expression network was filtered for OA genes identified in the meta-analysis, alongside their first-order interactors. Since the network comprised 31 disconnected components, components with fewer than 10 nodes were filtered out.

### 4.6. Multi-Source Information Network Construction and Gene Rankings

Selected OA multi-source data were integrated by extending a previously proposed methodology [10]. In total, three different sources of information were utilised: validated miRNA targets (from miRTarBase [7]), the co-expression network, and OA-associated SNPs. These individual networks were used to produce an MI network that is based on the weighted sum of the pairwise weighted edge vectors (for each pair of genes), and node-specific information from different sources was used to produce the weighted sum of the nodal score. An overview of this methodology is depicted in Figure 3 and a detailed description can be found in Appendix A.

### 4.7. Confirmation of Results with Mass Spectrometry Analysis

Results of the MI network were overlapped with label-free mass spectrometry proteomics data of human OA articular cartilage compared to healthy controls (manuscript in preparation). The mass spectrometry data have been deposited to the ProteomeXchange Consortium via the PRIDE [47] partner repository with the dataset identifier PXD029116 and 10.6019/PXD029116.

## 5. Conclusions

OA is a progressive and debilitating disease and the most common cause of chronic disability in adults. This study identifies 33 dysregulated miRNAs in human OA cartilage which may present as good candidates for replacement or inhibition therapy, as well as 207 differentially expressed mRNAs. Results of IPA and overlap analyses suggest a strong association between OA and senescence, corroborating the idea that the accumulation of senescent cells in cartilage contributes to the ECM degradation characteristic of OA. The MI network approach used in this study to integrate multi-source data may help to uncover important and novel genes involved in OA. Experimental confirmation of our bioinformatic analyses, using mass spectrometry data, revealed 32 proteins that are significantly differentially expressed in human OA cartilage. By using a range of bioinformatic methods, this study enabled the ranking, prioritisation, and experimental confirmation of novel OA-associated genes and their encoded proteins. Ultimately, this will allow for future research to focus on genes that may be of higher importance to OA pathogenesis and assess their suitability as drug targets or disease biomarkers. This is particularly important given that pain management and total joint replacement procedures are the only current treatment options for the disease.

## Figures and Tables

**Figure 1 ijms-23-04395-f001:**
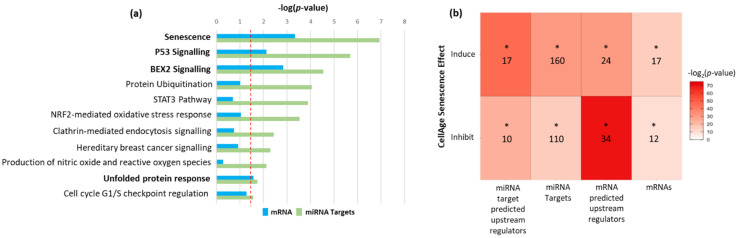
(**a**) The 12 canonical pathways, determined by Ingenuity Pathway Analysis (IPA), that were enriched for both the list of predicted miRNA target genes (green) and list of dysregulated mRNAs (blue) identified from the meta-analysis. Pathways significantly enriched in both miRNA targets and mRNAs are shown in bold. (**b**) A heatmap showing the overlap between miRNA target genes, mRNAs, and their predicted upstream regulators, with genes that have been shown in vitro to either induce or inhibit cellular senescence (CS). Numbers of overlapped genes are indicated in each cell. * *p* < 0.01 Fisher’s exact test with Benjamini–Hochberg false discovery rate correction.

**Figure 2 ijms-23-04395-f002:**
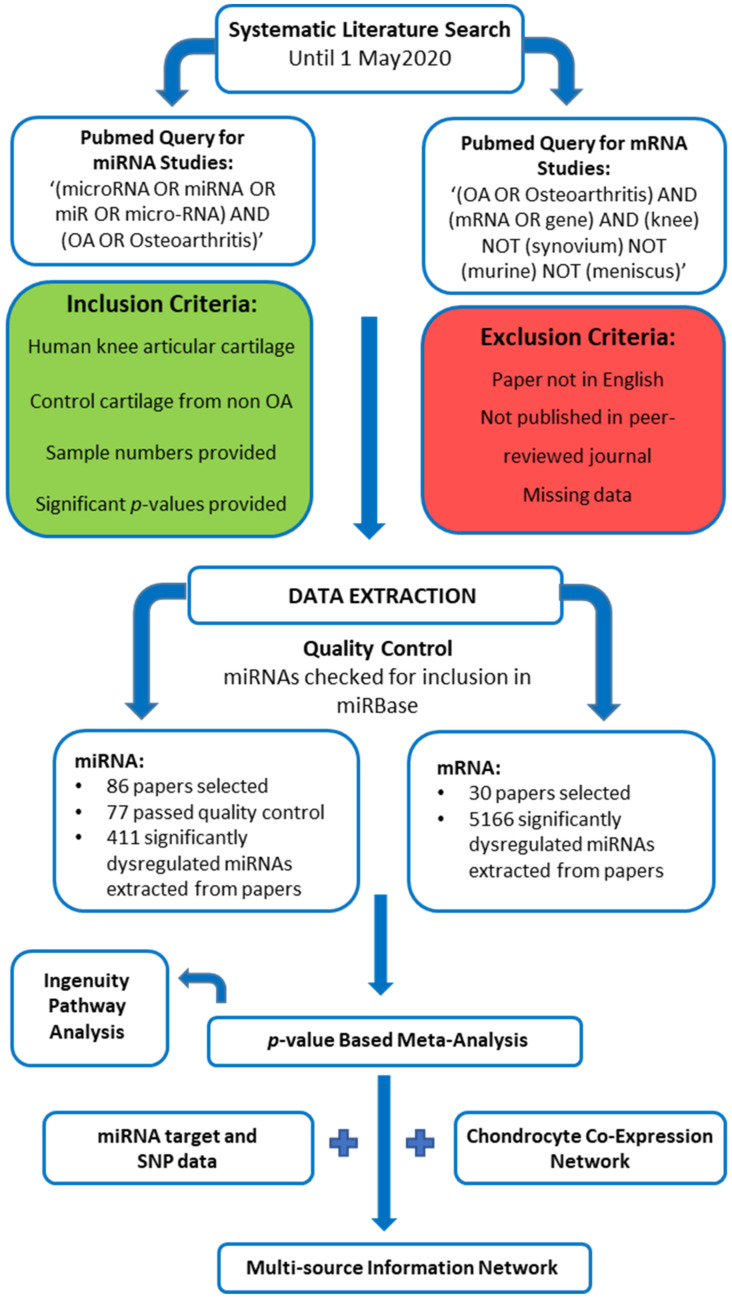
A schematic overview of the study design.

**Figure 3 ijms-23-04395-f003:**
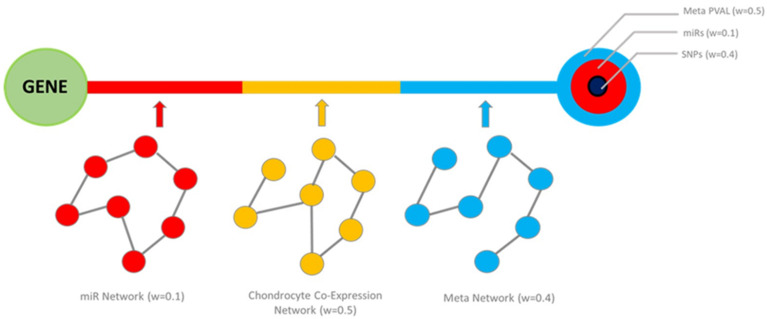
A depiction of the multi-omic integration method used to create the MI network of genes involved in OA. The circles represent the ‘nodes’ and the connecting line represents the ‘edge’. ‘w’ refers to the weight given to each source of data used to calculate the node and edge-specific scores.

**Table 1 ijms-23-04395-t001:** All miRNAs and the 20 most significant mRNAs found to be dysregulated in knee articular cartilage of OA patients compared to controls in three or more independent studies.

miRNA/mRNA	Summed Z-Score	*p*-Value
*DDIT4*	−10.63912988	9.80 × 10^−27^
*TXNIP*	−9.159840611	2.60 × 10^−20^
*RPL23AP1*	−8.608480816	3.70 × 10^−18^
*C10orf10*	−8.302559066	5.09 × 10^−17^
*ANG*	8.070197563	3.51 × 10^−16^
*APOD*	−7.769000282	3.96 × 10^−15^
*GPX3*	−7.67384789	8.35 × 10^−15^
*CEBPD*	−7.448716407	4.71 × 10^−14^
*DLX5*	−7.351922828	9.77 × 10^−14^
*HOXA5*	−7.319143505	1.25 × 10^−13^
*GDF15*	−7.277596172	1.70 × 10^−13^
*PDK4*	−7.26107927	1.92 × 10^−13^
*CISH*	−7.150042437	4.34 × 10^−13^
*SCNN1A*	6.872376244	3.16 × 10^−12^
*RND1*	−6.869197998	3.23 × 10^−12^
*CSNK2A2*	−6.803140204	5.12 × 10^−12^
*KLF15*	−6.746554293	7.57 × 10^−12^
*DCXR*	−6.740626982	7.89 × 10^−12^
miR−149	−4.31654501	7.92 × 10^−6^
miR−150−5p	−3.679922531	1.17 × 10^−4^
miR−140	−3.628394273	1.43 × 10^−4^
miR−140−5p	−3.598588637	1.60 × 10^−4^
miR−424−3p	−3.396430809	3.41 × 10^−4^
miR−26a	−3.099660248	9.69 × 10^−4^

**Table 2 ijms-23-04395-t002:** The top 20 OA-associated genes identified from the MI network. Genes are ranked by total gene score—based on both node and edge aggregated score.

Gene	Total Gene Score
*APOD*	0.769747271
*PDK4*	0.759162932
*HNRNPH3*	0.754176306
*G0S2*	0.727109931
*CISH*	0.726721385
*GDF15*	0.709522833
*TXNIP*	0.706369071
*TIMP4*	0.700535338
*DDIT4*	0.691347206
*CALCA*	0.690417491
*RPL5*	0.654961725
*SCNN1A*	0.652784123
*UQCR10*	0.645728211
*GPX3*	0.645571939
*PLIN5*	0.644970423
*PRLR*	0.62678813
*CHI3L1*	0.621938649
*RND1*	0.615125789
*LIF*	0.613030601
*HOXA5*	0.612272604

**Table 3 ijms-23-04395-t003:** OA-associated genes identified both at the mRNA level from the MI network analysis and at the protein level by mass spectrometry. Negative and positive fold changes (FC) show down- and upregulation in OA cartilage vs. control, respectively.

Gene Name	Integrated Network Ranking	Included in Meta Analysis	Included in CellAge Database	Mass Spec OA Cartilage vs. Control		
				Superficial Zone FC	Middle Zone FC	Deep Zone FC
*APOD*	1	TRUE	FALSE	−1.259817494	−2.410027844	−2.146022337
*GPX3*	14	TRUE	FALSE		−1.703092803	
*SERPINA1*	35	TRUE	FALSE	−3.737089031	−2.102085205	
*PARK7*	40	TRUE	TRUE	−2.178129473		
*ACAN*	94	TRUE	FALSE	−1.918704064		
*TRPV4*	104	TRUE	FALSE	−1.49270102		1.776227668
*LDHA*	105	TRUE	FALSE	−1.375252605		1.052821799
*FBN1*	133	FALSE	FALSE		0.745264827	1.304169331
*PPIB* ^1^	283	FALSE	TRUE	1.069319432		
*GSTP1*	341	FALSE	FALSE	−1.643744729	−1.582648578	
*COL1A1*	402	FALSE	FALSE	1.49156546		
*CA1*	454	FALSE	FALSE	−3.386795108		
*POSTN*	467	FALSE	FALSE	2.640518847		
*HP*	595	FALSE	FALSE	−4.004374124		
*ANXA2*	609	FALSE	FALSE			0.832045847
*RHOC*	711	FALSE	FALSE	−1.796760511		
*LBP*	766	FALSE	FALSE	−2.640551511		
*TGFBI*	772	FALSE	TRUE	1.923736073		
*RPS27A*	857	FALSE	FALSE	−1.278041904		
*PRG4*	997	FALSE	FALSE		−1.640433802	
*PGK1*	1033	FALSE	FALSE	−1.104244463		
*ASS1* ^1^	1099	FALSE	FALSE	−2.641457504		
*PRDX1*	1144	FALSE	FALSE	−1.120350112		1.10599408
*LDHB* ^1^	1153	FALSE	FALSE	−1.358199289		
*COL12A1*	1203	FALSE	FALSE		1.574786097	
*ANXA5*	1204	FALSE	TRUE			0.881830367
*TPI1* ^1^	1214	FALSE	FALSE	−1.555403271		
*PKM*	1267	FALSE	TRUE			1.12886159
*MATN3*	1295	FALSE	FALSE		2.766971378	
*ARPC4*-*TTLL3*^1^	1371	FALSE	FALSE			1.386654494
*A2M*	1542	FALSE	FALSE	−2.854802653		
*PRDX2*	1574	FALSE	FALSE	−3.152028639		

^1^ Proteins that have never been associated with OA previously.

## Data Availability

The mass spectrometry data used in this study have been deposited to the ProteomeXchange Consortium via the PRIDE partner repository with the dataset identifier PXD029116 and 10.6019/PXD029116.

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
