# Peer review of "Identifying Novel Osteoarthritis-Associated Genes in Human Cartilage Using a Systematic Meta-Analysis and a Multi-Source Integrated Network"

_ijms, 2022, doi:10.3390/ijms23084395_

Round 1

Reviewer 1 Report

Emily Shorter et al. performed outstanding research based on systemic meta-analysis and multi-source integrated network analysis of osteoarthritis-associated genes and proteins. They identified 6 significantly dysregulated miRNAs and 207 mRNAs by p-value-based meta-analysis and submitted these to Ingenuity Pathway Analysis. The targets could fit four common pathways, out of which the senescence is the most important. The CellAge genes inducing cellular senescence showed a 43% overlap with predicted upstream regulators of differentially expressed mRNAs. An integrated multi-source information (MI) network was constructed with 1685 genes and filtered for dysregulated osteoarthritic proteins. Comparison of MI and mass-spectrometry results revealed 32 common proteins, with 5 new osteoarthritis candidate markers: PPIB, ASS1, LDHB, TPI1, and ARPC4-TTLL3.

These are comprehensive results obtained with a complex methodology and novel aspects worthwhile to be published. The implication of the senescence-associated secretory phenotype signatures in low-grade cartilage inflammation and inflammaging should be further exploited in future research. Before the publication of the manuscript, I suggest several improvements:

  1. One of the essential results is obtained in a comparative analysis of the MI network and MS data, highlighting 32 differentially regulated proteins. In some cases, differential expressions were described in the superficial/middle/deep cartilage zones. These results should be introduced in the abstract.
  2. For a better understanding, the manuscript would benefit from introducing a study design graph.
  3. The authors found five new protein biomarker candidates for osteoarthritis. However, in Discussions, they commented only on PPIB and LDHB. Some comments about the potential role of ASS1, TPI1, and ARPC4-TTLL3 are lacking.
  4. A few grammatical misspellings should be revised, like:

“osteoarthri-tis-associated”, “system-atic”,  “en-riched”,  “re-vealed”

in the Abstract, or

L181-182: “co-expression network allows identification of genes that tend 181 to show a coordinated expression pattern across a group of samples11 in this”. 

Author Response

Thank you for your comments, it's really appreciated! Please find our response to your suggestions below:

1. One of the essential results is obtained in a comparative analysis of the MI network and MS data, highlighting 32 differentially regulated proteins. In some cases, differential expressions were described in the superficial/middle/deep cartilage zones. These results should be introduced in the abstract.

I have added a brief sentence introducing the three zones of the cartilage in the abstract.

2. For a better understanding, the manuscript would benefit from introducing a study design graph.

I have added a flow chart of the methodology at the beginning of the methods section. Hopefully this makes the study design a bit clearer. 

3. The authors found five new protein biomarker candidates for osteoarthritis. However, in Discussions, they commented only on PPIB and LDHB. Some comments about the potential role of ASS1, TPI1, and ARPC4-TTLL3 are lacking.

There was very little research on these 3 genes which is why I didn't mention them in the original manuscript. You're right though, they should be discussed! I have added a sentence on what is known about the role of each of the genes in the discussion. I hope this suffices!

4. A few grammatical misspellings.

Thanks for bringing our attention to this - I think pasting the manuscript onto the template caused some hyphens to be added in between words. I have removed these now and fixed any spelling errors I could find.

Thanks again for your help with reviewing this paper! 

Reviewer 2 Report

The authors shouldered a big task of summarising and synthesising the information presented in multiple other papers - a strong point of this study over other met-analyses is that it was able to convincingly reveal the differences in importance of the possible contributing pathways - something that is awfully difficult in any single study. I have only one comment that is a limitation of the study not mentioned in the text: "The p-value 72 based meta-analysis identified 6 miRNAs and 207 mRNAs as being significantly dysreg- 73 ulated in OA cartilage compared to healthy tissue in 3 or more independent studies." The choice of what type of RNA is being studies by any individual researcher is arbitrary and has nothing to do with the importance of the given molecule in the pathogenesis of OA. If we use the criterion "minimum 3 independent studies found it" it reflects the preference of researchers just as well as the solidity of the findings. One can imagine that an obscure but rather critical mRNA that is investigated by only one group worldwide does not make the cut in the present analysis despite its importance. This selection bias shield be discussed in the revised manuscript. 

Author Response

'The choice of what type of RNA is being studied by any individual researcher is arbitrary and has nothing to do with the importance of the given molecule in the pathogenesis of OA. If we use the criterion "minimum 3 independent studies found it" it reflects the preference of researchers just as well as the solidity of the findings'.

You're absolutely right! I tried to touch on this in the discussion, but I don't think it was clear enough. I have added a sentence to make it clearer. Here is the part of the discussion where I mention this bias in methodology:

'Moreover, research has suggested that there are reporting biases of differential gene expression in literature, including: preferential reporting of overexpressed rather than under-expressed genes as well as genes that are popular in the biomedical literature at large [33]. As such, a critical mRNA that is investigated by only one group worldwide may not make the cut in the present analysis despite its potential importance to the disease pathogenesis. This bias is evident in the results of this study. For example, miR-140 is probably the most researched and established miRNA to date in terms of its relation to OA [34, 35]. As its dysregulation has been very well classified, research will often include it as a positive control. This is reflected in the results of this meta-analysis, where miR-140, miR-140-3p, and miR-140-5p were all found to be significantly dysregulated. '

I hope this provides some clarification. Thanks so much for your help with reviewing this paper, it's much appreciated!